# Estimating the Heavy Metal Contents in Entisols from a Mining Area Based on Improved Spectral Indices and Catboost

**DOI:** 10.3390/s24051492

**Published:** 2024-02-25

**Authors:** Pingjie Fu, Jiawei Zhang, Zhaoxian Yuan, Jianfei Feng, Yuxuan Zhang, Fei Meng, Shubin Zhou

**Affiliations:** 1School of Surveying and Geo-Informatics, Shandong Jianzhu University, Jinan 250101, China; fupjrs@126.com (P.F.); 2021165111@stu.sdjzu.edu.cn (J.F.); zyx970614@163.com (Y.Z.); lzhmf@sdjzu.edu.cn (F.M.); 2College of Geodesy and Geomatics, Shandong University of Science and Technology, Qingdao 266590, China; 3Institute of Resource and Environmental Engineering, Hebei Geo University, Shijiazhuang 050031, China; sdyzx86@126.com; 4School of Earth Sciences and Engineering, Sun Yat-sen University, Zhuhai 519000, China; 5School of Natural Sciences, Faculty of Science and Engineering, Macquarie University, Sydney, NSW 2109, Australia

**Keywords:** entisols, hyperspectral remote sensing, heavy metals, spectral index, Catboost

## Abstract

In the study of the inversion of soil multi-species heavy metal element concentrations using hyperspectral techniques, the selection of feature bands is very important. However, interactions among soil elements can lead to redundancy and instability of spectral features. In this study, heavy metal elements (Pb, Zn, Mn, and As) in entisols around a mining area in Harbin, Heilongjiang Province, China, were studied. To optimise the combination of spectral indices and their weights, radar plots of characteristic-band Pearson coefficients (RCBP) were used to screen three-band spectral index combinations of Pb, Zn, Mn, and As elements, while the Catboost algorithm was used to invert the concentrations of each element. The correlations of Fe with the four heavy metals were analysed from both concentration and characteristic band perspectives, while the effect of spectral inversion was further evaluated via spatial analysis. It was found that the regression model for the inversion of the Zn elemental concentration based on the optimised spectral index combinations had the best fit, with *R*^2^ = 0.8786 for the test set, followed by Mn (*R*^2^ = 0.8576), As (*R*^2^ = 0.7916), and Pb (*R*^2^ = 0.6022). As far as the characteristic bands are concerned, the best correlations of Fe with the Pb, Zn, Mn and As elements were 0.837, 0.711, 0.542 and 0.303, respectively. The spatial distribution and correlation of the spectral inversion concentrations of the As and Mn elements with the measured concentrations were consistent, and there were some differences in the results for Zn and Pb. Therefore, hyperspectral techniques and analysis of Fe elements have potential applications in the inversion of entisols heavy metal concentrations and can improve the quality monitoring efficiency of these soils.

## 1. Introduction

Entisols occur in every part of the world and are often affected by serious soil erosion or extensive agricultural activities [1]. Therefore, the detection of their quality is of great practical importance. With the increasing human demand for mineral resources, the accumulation of mining waste and slag has not only degraded land resources but has also degraded the soil ecology around mining areas via heavy metal pollution [2]. Hyperspectral remote sensing technology provides an efficient approach to the in situ monitoring of soil heavy metal contents as it provides information with good timeliness, spectral resolution, and measurement range [3,4,5].

Malley and Williams were the first to employ hyperspectral remote sensing technology to quantitatively invert the heavy metal concentrations in lake sediments, achieving good prediction accuracy [6]. Since then, many scholars have used hyperspectral remote sensing technology for the determination of heavy metal contents in soil and have made great progress [7,8,9,10]. Studies on the direct inversion of heavy metal concentrations in soil using hyperspectral remote sensing have found that derivative processing achieves better results than spectral data preprocessing with S-G convolution smoothing, inverse logarithm, multiple scattering correction, and envelope removal [11,12,13]. Basically, the characteristic bands of target elements are obtained by correlation analysis of the target heavy metal elements with spectral transform data and are then used for the estimation of elemental concentrations [14,15,16,17]. Later, other studies found that the construction of multi-band spectral indices using the characteristic bands of target elements could improve the inversion of soil heavy metal concentrations [18,19]. Peng et al. conducted a Pearson’s correlation analysis on the spectra and soil heavy metal concentration, and a total of 13 feature bands were determined for the inversion model [20]. Sawut et al. have confirmed that the dual-band spectral indices could be applied to the estimation of the content of As [19]. The theoretical basis for using hyperspectral remote sensing technology in soil heavy metal research is being gradually improved [21]. As far as inversion models are concerned, classical models of univariate and multivariate statistical analysis were mostly used in early studies [22,23,24]. However, given the influence of soil mineral composition, particle size, water content, and organic matter concentration on soil spectra, machine learning algorithms have been increasingly used in the field of quantitative estimation to improve the retrieval efficiency of heavy metal concentrations [25,26]. In addition, solar radiation conditions may affect the hyperspectral measurement. For example, clouds may change the illumination regime from directed illumination to diffuse illumination conditions and affect hyperspectral measurements [27]. By using statistical models and machine learning algorithms to study the correlation between soil heavy metal element concentrations and hyperspectral reflectance data, a commendable level of accuracy in estimating soil heavy metal content can also be achieved based on laboratory hyperspectral data [28], airborne hyperspectral imagery [29], and spaceborne hyperspectral imagery [28,30]. The airborne hyperspectral imagery can offer finer details [29], laboratory hyperspectral data is less susceptible to interference and can provide higher accuracy [28], and spaceborne hyperspectral imagery is more suitable for cost-effective large-scale pollution assessments [28,30]. However, regarding the problem of redundant and unstable spectral features due to interaction effects among soil elements, the optimisation of different elemental feature bands or spectral indices is not often considered in the inversion of heavy metal concentrations. In this study, radar plots of characteristic-band Pearson coefficients (RCBP) were used to screen combinations of the three-band spectral indices of different heavy metals. Combined with the Catboost algorithm, which provides automatic optimisation of parameter weights, the present research estimated heavy metal concentrations (Mn, Zn, As, and Pb) in soils using hyperspectral remote sensing.

Previous studies indicate that indirect measurements of heavy metal concentrations using hyperspectral remote sensing techniques are possible and that the relationships between heavy metal concentrations and soil components (e.g., clay minerals, carbonate minerals, iron oxides, and organic matter) are well established. These components have a strong absorption effect on metal cations and are key factors influencing soil spectral morphological characteristics. This means that the characteristic bands of the corresponding soil spectral curve response are easy to identify. Strong correlations between heavy metals and soil components may lead to better measurement [31,32,33,34]. Due to the complex composition of clay and carbonate minerals, most studies have constructed models for estimating indirect inversion based on the concentrations of Fe elements or organic matter and their characteristic bands [3,9,35,36]. Xia et al. analysed the correlation between eight types of heavy metal elements and organic matter concentrations, compared the locations of sensitive bands for different heavy metal elements and organic matter, and developed a partial least squares regression model for the inversion of heavy metal concentrations [37]. Cheng et al. studied soils in suburban Wuhan, Hubei Province, China, using hyperspectral techniques and found that the spectral estimates of Cd concentrations were strongly correlated with organic matter, while Cr and As were closely correlated with Fe, but the concentrations of Pb, Zn, and Cu were weakly correlated with either organic matter or Fe, leading to poor estimation results [35]. Shen et al. analysed the relationship between Fe and heavy metal concentrations and predicted Fe concentrations using transformed spectral data [38]. They then predicted heavy metal Cu concentrations based on the predicted Fe concentrations. However, there are relatively few such studies in entisols. Therefore, we aimed to analyse the correlations between Fe and the concentrations and characteristic bands of four heavy metals (Pb, As, Zn, and Mn. We used Pearson coefficients and radar plots to assess the importance of Fe elements to the estimation of their concentrations in entisols.

In summary, the redundancy and instability of the spectral features of the heavy metal elements in the entisols necessitate the development of a concentration inversion model to optimise the combination of spectral indices and parameter weights. Furthermore, the effect of Fe concentration on the inversion of other heavy metal elements content in entisols (e.g., Mn, Zn, As, and Pb) based on the hyperspectral method requires further investigation. In this study, we developed the inversion model of heavy metal concentrations (Mn, Fe, Zn, As, and Pb) by collecting spectral curves and composition data from entisols in a metal mining-impacted area in Harbin, China, aiming to demonstrate the feasibility of monitoring entisols quality using hyperspectral remote sensing technology. This study is divided into three parts:(1)The three-band spectral indices with good correlations with concentrations of Mn, Zn, As, and Pb were extracted. The combinations of spectral indices were screened by RCBP to invert the concentrations of each heavy metal using the Catboost algorithm.(2)The correlations between Fe concentration and elemental concentrations (and characteristic bands) of Mn, Zn, As, and Pb were established through Pearson coefficient analysis and radar plotting.(3)The spatial distribution and correlation of heavy metal elements in entisols in a metal mining-impacted area were determined using the spectral inversion concentration data of Mn, Zn, As, and Pb in soils.

## 2. Materials and Methods

### 2.1. Study Area and Dataset

The study area, approximately 8500 m^2^ in size, was situated in a maise field (Figure 1), located 1.5 km northwest of the Bailing Cu-Zn deposit in Xiaoling County, A’cheng District, Harbin City, Heilongjiang Province, China. The deposit is a skarn deposit, and the ore mineralogy is dominated by magnetite, sphalerite, arsenopyrite, and chalcopyrite, with minor pyrite and hematite occurrences [39]. The average ore grades in the main ore body for Fe, Cu, Zn, and As are 28.45%, 0.5%, 1.88%, and 5.51%, respectively [39]. The exploration of the deposit was started in the 1980s and has been exploited since 2009 [39]. Soils of the maise field are Entisols per the US Soil Taxonomy [40]. The research area has a continental climate (Dwa, according to the Koppen-Geiger climate classification), with dry and cold winters and hot and humid summers [41]. Precipitation is mainly concentrated during summer (June to September), and the annual precipitation is approximately 518 mm [39]. A small stream with a width of 1–2 m (during the dry season) passes by the deposit and is located along the west side of the studied maise field. The wastewater from the deposit forms a branch of the stream. The studied maise field is located in a low-lying area, where monsoons flood during the rainy season (June to September), likely submerging the overbank on the west side of the maise field.

A total of 95 soil samples were collected from the maise field at a depth of 0–20 cm, following a roughly 10 × 10 m grid pattern. The collected soil samples were then oven-dried at 65 °C for 72 h, ground using a rubber mallet, and sieved using a 0.18 mm mesh. The sieved soil sample then underwent ex situ pXRF analysis using a Niton XL3t 950 (Thermo-Fisher Scientific, Waltham, MA, USA) pXRF analyser. The sieved soil samples were collected in a homemade sample cup with a diameter of 23 mm and a height of 25 mm [42]. The sample cup was sealed by polyethylene films (tens of mm, with nearly no compositional interference and high X-ray transmission rate), which have been used in previous studies for protecting the pXRF detection window [43,44]. The sample cup was placed in a test stand and then underwent PXRF analysis (aperture up) using line power at 220 VAC [45]. The dwell time for pXRF analysis was 90 s, and eight replicate measurements were conducted for each soil sample. In order to monitor drift, insertion of certified reference material (CRM) 180–661 was conducted at regular intervals. The recoveries for As, Pb, and Fe are 87–91%, 92–99%, and 91–94%, respectively. 

The ex situ pXRF data quality was also examined by the ICP-MS and ICP-AES method conducted at ALS Minerals (Guangzhou, China) per Chinese Standard SY/T 6404-2018 [46]. To this end, 0.25 g samples were first digested using a perchloric and nitric acid mixture (As3+ was oxidated to As5+, preventing As evaporation). Then, hydrofluoric acid was added, and the digestion vessels were heated using an EG 20A-PLUS hotplate produced by LabTech. Then, hydrochloric acid was added to the residues (after heating evaporation) and brought to a constant volume. Lastly, the prepared solutions underwent ICP-MS and ICP-AES analysis using an Aglient 7900 instrument. Arsenic, Mn, Pb, and Zn were determined using the ICP-MS method, and Fe was determined using the ICP-AES method. The LODs of the ICP-MS method for As, Mn, Pb, and Zn are 0.2 mg kg^−1^, 5 mg kg^−1^, 0.5 mg kg^−1^, and 2 mg kg^−1^, respectively. The LOD for Fe of the ICP-AES method is 100 mg kg^−1^. The accuracy of ICP-MS and ICP-AES measurement was confirmed by measuring three certified reference materials for soils (GBM908-10 supplied by Geostats Pty Ltd., O’Connor, WA, Australia, MRGeo08 supplied by ALS Minerals, Vancouver, BC, Canada, OREAS-25a supplied by Ore Research and Exploration Pty Ltd., Bayswater North, VIC, Australia) at regular intervals. The recovery percentages (ICP determined/CRM reported) for As, Fe, Mn, Pb, and Zn are 98–112%, 100–103%, 99–103%, 94–100%, and 97–105%, respectively. Furthermore, replicate ICP-MS/ICP-AES measurements were conducted on random samples to observe the standard deviations of the ICP-MS measurements. Standard deviations (and relative standard deviations) of replicate measurements can reflect the uncertainty of ICP-MS analysis. The average relative standard deviations of replicate ICP-MS measurements were 1.22%, 0.76%, 1.65%, 2.97%, and 5.61% for As, Fe, Mn, Pb, and Zn, respectively. The ex situ pXRF determined Mn, Fe, Zn, As, and Pb concentrations were highly correlated with those determined by ICP-MS. The *R*^2^ values for Mn, Fe, Zn, As, and Pb were 0.99, 0.96, 0.97, 0.99, and 0.93, respectively, indicating a good pXRF data quality. Therefore, the Mn, Fe, Zn, As, and Pb concentrations determined by ex situ pXRF were quite reliable and used for further comparison. 

The dried and sieved soil samples were measured by using a FieldSpec 4 Hi-Res portable geophysical spectrometer produced by ASD (Analytical Spectral Devices) corporation in the USA with a wavelength of 350–2500 nm in midnoon (9:30 a.m.–14:30 p.m.) on a clear and sunny day in summer. The 95 soil samples (dried and sieved) were laid in 10 × 10 cm square size flat on a black cloth to reduce the reflection interference. The spectrometer’s fibre probe with a view angle of 25° was positioned approximately 18 cm perpendicular to the surface of the sample. Two people with dark clothes were involved in the scanning process, with one operating a laptop connecting FieldSpec 4 Hi-Res portable ASD geophysical spectrometer and another preparing the samples for measurement. Each soil sample underwent 10 replicate measurements to reduce errors, and the average of the spectra was obtained for the following study.

### 2.2. Methods

Figure 2 shows a flowchart of the whole procedure, which included three steps: (i) first-order derivative pre-processing of spectral data and acquisition of Pb, As, Zn, Mn and Fe elemental concentrations. (ii) Extraction of three-band spectral indices of Pb, Zn, Mn and As concentrations using a three-dimensional slicing method and screening of three-band spectral index combinations based on radar plots of the characteristic band Pearson coefficients (RCBP) using MATLAB R2017b and Excel 2021. (iii) Inversion of the heavy metal concentrations of Pb, As, Zn, and Mn based on the preferred combination of spectral indices and Catboost model, followed by the correlation of Fe with the characteristic bands and concentrations of the four heavy metal elements using Pearson coefficients and radar plots. (iv) Spatial analysis of the concentrations of the four heavy metals based on the concentrations of spectral inversion.

#### 2.2.1. Preferred Combination of Spectral Indices Based on RCBP

Analysis of the existing results revealed that among the soil spectral pre-processing methods, excellent results were achieved by the derivative processing technique, which effectively eliminated the environmental background interference [11]. The first-order derivative data of the spectra were used to construct highly accurate three-band spectral indices with the inversion of soil heavy metal concentrations [18]. First, the collected soil spectra were processed for first-order derivatives. Then, to make full use of the spectral information, three combined forms of three-band spectral indices (SI) were constructed: SI1: (Ri − Rj)/(Ri + Rk), SI2: (Ri − Rj) + (Rk − Rj) and SI3: Ri/(Rj + Rk) (Figure 3a). The spectral indices that correlated well with the concentrations of Pb, As, Zn, and Mn were retained by traversing all possible triple-band combinations over the entire band range (Figure 3b).

RCBP was used to screen the combination of three-band spectral indices, which included the following steps. Based on the spectral indices of the four heavy metals that were retained (Figure 3b), the Pearson correlation coefficients of each element concentration and its three-band spectral index were obtained. The spectral indices were ranked from the largest to smallest correlation coefficient (Figure 3b), and then the spectral index combinations were determined to obtain the characteristic bands (Figure 3c). Pb, As, Zn and Mn were taken as the target elements in turn, and the correlation coefficients between the characteristic bands of the target elements and those of other elements were calculated under the condition of maintaining the same number of spectral index combinations (Figure 3d). The RCBPs between the target element and the other three elements were drawn separately (Figure 3e), and the spectral index combinations of the target element were further screened according to the principle of minimum correlation (Figure 3f). These spectral index combinations were used to construct inversion models of the concentrations of Pb, As, Zn and Mn in the soil. In Figure 3, Pearson coefficient 1 is the correlation between heavy metal element concentrations and spectral indexes. Pearson coefficient 2 is the correlation coefficient between the characteristic bands of the target element and those of other elements calculated according to the same number of spectral index combinations. The target element was cycled between As, Pb, Mn and Zn, with As used here as an example. The first index combination is the SI with the best correlation with the target element, the Top 2 index combinations are the first two SI with good correlation with the target element, and so on. The top 5 index combinations are the top five SI with good correlation with the target element.

#### 2.2.2. Catboost Determination of Heavy Metal Concentrations

Catboost is a new open-source machine learning library proposed by Yandex in 2017, which consists of Categorical and Boosting components. It is a gradient-boosted decision trees (GBDT) framework based on a symmetric decision tree algorithm. It not only has fewer parameters and high accuracy but also supports categorical variables. The efficient and reasonable handling of categorical features is also its main advantage. The Catboost model overcomes the gradient bias and prediction offset problems of the traditional Boosting framework, which reduces the occurrence of overfitting and improves the accuracy and generalizability of the model. In the model training process, Catboost uses a serial approach to integrate multiple base learners. The training sample set remains the same in each round, and the sample weights are continuously updated by the learning results of the previous round, thus gradually reducing the bias caused by noise points. There is a dependency between the multiple weak learners generated by the training, and the final result is obtained by weighting the regression values of all the weak learners (Figure 4). The construction, training, and testing of the Catboost is completed in an integrated development environment based on Python and Anaconda.

The coefficient of determination (*R*^2^) and the root-mean-square error (*RMSE*) were chosen to assess the stability and accuracy of the models. The *R*^2^ indicates the stability of the model, with values closer to 1 indicating greater stability. The *RMSE* indicates the accuracy of the model, with smaller values indicating greater accuracy. The formulae for calculating the *R*^2^ and *RMSE* are as follows.
$$\begin{array}{l} R^{2}=\sum_{i=1}^{n}{\left({\hat{y}}_{i}-\bar{y}\right)}^{2}/\sum_{i=1}^{n}{\left(y_{i}-\bar{y}\right)}^{2}\\ RMSE=\sqrt{\frac{1}{n}\sum_{i=1}^{n}{\left({\hat{y}}_{i}-y_{i}\right)}^{2}} \end{array}$$
where ${\hat{y}}_{i}$ is the predicted values, $\bar{y}$ is the average of the observed values, $y_{i}$ is the observed values, and *n* is the number of predicted/observed values. The ratio of training set to test set in this study is 8:2.

#### 2.2.3. Spatial Analysis

The accuracy of the inversion results was evaluated from three aspects: spatial distribution, spatial correlation, and spatial clustering. The spatial correlation and clustering of the four heavy metal elements were analysed. Spatial interpolation methods are widely used in resource and disaster management and environmental management. The inverse distance-weighted average interpolation (IDW) method is based on the theory of the first law of geography. This considers that a predicted point becomes less affected with distance from an observation point and is one of the more applied interpolation methods [47].

## 3. Results

### 3.1. Spectral Index Combination Preference and Heavy Metal Concentration Assessment

The spectral index combination optimisation is completed using the RCBP method, with the first step being the extraction of SIs. The spectra of all soil samples were processed for first-order derivatives, and then the correlations between the first-order derivative data and the measured concentrations of Pb, Zn, Mn, As, and Fe elements were analysed. The spectral indices with good correlations with each element concentration were extracted. The top five groups of spectral indices with good correlations with the five elemental contents were obtained separately (Table 1). The spectral indices were found to be SI2 and SI3 for Pb concentration, SI2 for Zn concentration, SI1 and SI3 for both Mn and Fe, and SI1 for As. According to the correlation coefficient r, the As concentration showed the best correlation with the spectral indexes, followed by Zn, Mn and Pb.

The latter is to optimise the spectral characteristic bands of each element. To reduce the redundancy of spectral features due to the existence of interactions among soil elements, the spectral indices of Pb, As, Zn and Mn extracted from Table 1 were combined according to the ranking of correlation coefficients (taking Pb as an example, the optimal three-band spectral index is the first index in the table; i.e., ① (R_FD1278_ − R_FD622_) + (R_FD1530_ − R_FD622_). The top two spectral index combinations are ① (R_FD1278_ − R_FD622_) + (R_FD1530_ − R_FD622_) and ② (R_FD466_ − R_FD622_) + (R_FD1530_ − R_FD622_). The top three spectral indices were combined as ① (R_FD1278_ − R_FD622_) + (R_FD1530_ – R_FD622_), ② (R_FD466_ − R_FD622_) + (R_FD1530_ − R_FD622_) and ③ (R_FD450_ − R_FD622_) + (R_FD1530_ − R_FD622_). The top four spectral index combinations are ① ② ③ ④ and the top five spectral index combinations are ① ② ③ ④ ⑤). With combinations of the same number of spectral indices, Pb, As, Zn and Mn were taken in turn as the target element, and the Pearson coefficients between the characteristic bands of the target elements and those of the other three elements were calculated and analysed. Based on the results, separate radar plots of the correlation coefficients between Pb, As, Zn and Mn were obtained (Figure 5). Figure 5 shows that the top three spectral index combinations were suitable for selection to ensure a small correlation between the characteristic bands of Pb and those of each other element; arsenic is the same case. The smallest correlation between the characteristic bands of Zn and those of each other element indicates the top two spectral index combinations and the same is true for Mn.

The top three well-correlated spectral index combinations of Pb and As were used to construct concentration inversion models with characteristic bands of 450, 466, 622, 1278 and 1530 nm for Pb and 622, 746, 930, 938, 1102, 1122 and 1274 nm for As. The top two well-correlated spectral index combinations of Zn and Mn were used for the inversion of elemental concentrations, with characteristic bands of 450, 622, 630, and 1230 nm for Zn and 1318, 1646, 1806, 2271, 2275, and 2383 nm for Mn. The Catboost algorithm was used to invert the elemental concentrations, and the accuracy of the test set used for each elemental concentration inversion model is shown in Figure 6. The regression model with the best fit was obtained as the inversion of Zn concentration, with an *R*^2^ of 0.8786 for the test set, followed by Mn (*R*^2^ = 0.8576), As (*R*^2^ = 0.7916) and Pb (*R*^2^ = 0.6022).

### 3.2. Correlations between Fe and Heavy Metal Elements in Terms of Concentrations and Spectra

The correlations between Fe and the Pb, As, Zn and Mn heavy metal elements in terms of concentration and characteristic bands were analysed using Pearson coefficients. The correlations between concentrations are shown in Figure 7.

The correlations of the characteristic bands of Fe with those of Pb, As, Zn and Mn were analysed using Pearson coefficients and radar plots (Figure 8). The spectral indices ① (in Table 1) of Pb have the best correlation with Fe, with a correlation coefficient of 0.837, followed by the ② and ④. The ③ and ⑤ have weaker correlations with Fe, with correlation coefficients of 0.276 and 0.131, respectively. The spectral indices ④ (in Table 1) of Mn showed the best correlation with Fe, with a correlation coefficient of 0.542, and the ②, ③, ①, and ⑤ showed weaker correlations with Fe, with correlation coefficients of 0.299, 0.289, 0.221 and 0.199, respectively. The spectral indices ④ (in Table 1) of Zn elements have the best correlation with Fe, with a correlation coefficient of 0.711, followed by the ⑤ and ④, while the ③ and ① have the weakest correlations with Fe, with correlation coefficients of 0.22 and 0.004, respectively. The spectral indices ① (in Table 1) for As correlate best with Fe, with a correlation coefficient of 0.303, followed by the ⑤, ③ and ②, and the ④ correlate least with Fe, with a correlation coefficient of 0.089. In summary, Fe has the best correlation with Pb in the 458, 654 and 1090 nm characteristic bands (r = 0.837). In the 458, 474, 654 and 1090 nm characteristic bands, Fe has the best correlation with Zn (r = 0.711). In the 458, 474, 522, 630, 654, 930, 1090, 1330 and 1694 nm characteristic bands, Fe has the highest correlation with Mn (r = 0.542). The correlations between Fe and As in the characteristic bands are low, with coefficients of r < 0.303.

### 3.3. Analysis of the Spatial Distribution of Soil Heavy Metal Concentrations Based on Spectral Inversion

#### 3.3.1. Spatial Distribution

To test the effect of spectral inversion, inverse distance-weighted interpolation of the inverse concentrations of Pb, As, Zn, and Mn at 95 sampling points was performed to obtain the spatial distribution of soil heavy metal contents (Figure 9). It can be seen from Figure 9 that after interpolation, the distributions of measured and spectral-predicted As and Mn values are in good agreement, with the concentrations of As being generally higher in the west and lower in the east, while the concentrations of Mn are generally higher in the west and north and lower in the central, east and south. The spatial distributions of spectral-predicted and measured concentrations of Pb and Zn are also relatively consistent; however, for Pb, the predicted concentrations are locally lower than the measured concentrations. For Zn, the predicted concentrations are locally higher than the measured concentrations.

#### 3.3.2. Spatial Correlation Analysis

Based on the measured and spectral-predicted concentrations, Moran’s I values were used to determine the global Moran’s I values for the four heavy metals (Pb, As, Zn, and Mn) in the soils of the study area (Table 2). The global Moran’s I are all >0, indicating that all four heavy metals show positive spatial correlations, i.e., larger (smaller) elemental contents are more likely to be clustered [48], but their correlations are low. The Moran’s I of the predicted and measured concentrations of As and Mn are similar, while those of the predicted concentrations of Pb and Zn are much smaller than those of the measured concentrations. The P-values of the measured concentrations of the four heavy metals are <0.05, and the Z-scores are >1.96, indicating that there is a certain spatial aggregation of each heavy metal element [49]. The P-values of the predicted concentrations of As and Mn are both <0.05, and the Z-scores are both >1.96, but the P-values of the predicted concentrations of Pb and Zn are both >0.05 with Z-scores < 1.96. This indicates that in terms of spatial correlation, the predictions of As and Mn concentrations are good, while those of Pb and Zn concentrations are not very satisfactory.

#### 3.3.3. Spatial Clustering Analysis

The global Moran’s I only tell us whether the space appears clustered, discrete, or randomly distributed, but not where it appears [50]. Therefore, further local Moran’s I analysis was conducted to determine where outliers or agglomerations occur. Figure 10 shows the spatial distribution characteristics of the aggregation types of the four heavy metals. The spatial clustering of the measured and predicted concentrations of As and Mn are in good agreement, with most points being insignificant, followed by a high-high aggregation and one point with an anomalously high value. The spatial clustering of the measured and predicted concentrations of Pb and Zn differed, with most points showing consistent insignificance. However, the distributions of high-high aggregation, low-low aggregation, high anomalies, and low anomalies of the predicted and measured concentrations of Pb differed significantly. The distributions of high-high aggregation and high anomalies of measured and predicted concentrations of Zn also differed significantly. This indicates that the predicted concentrations of As and Mn are better than those of Pb and Zn from the spatial clustering analysis.

## 4. Discussion

To retrieve the heavy metal concentrations in soils using hyperspectral technology, previous research mainly focused on the extraction of characteristic bands, comparison of spectral transformation methods, construction of spectral indexes, and comparison of inversion models; the main emphasis typically lies in comparing the accuracy of inversion algorithms. Malmir et al. developed the partial least square regression (PLSR) models using spectral reflectance of soil, and the models performed a good prediction of Cu and Zn concentrations in soil [8]. Zhou et al. found that the combination of the first derivative and the random forest model performed the highest inversion accuracy for the six heavy metals (Mn, Cu, Zn, Pb, Cr and Ni) [36]. Pyo et al. suggested that the characteristic bands of Al, Cu, As, Zn, Pb, Cd, Ni, Hg, and Cr partly overlap [25]. For the inversion of Cu in soils, the indirect inversion using Fe concentrations based on the backpropagation neural network (BPNN) model performed higher predictive accuracy than direct inversion using spectral characteristic bands of Cu [38]. Bian et al. showed that the BPNN model was better than the PLSR model in retrieving Cu, Zn and Pb concentrations, while the optimal spectral transformation methods for these elements are different [3]. Normally, the predictive accuracy of inversion models could be reflected by the comparison between measured and predicted concentrations (and the derived spatial distribution maps) [19,38]. The selection of the optimal combination of spectral parameters and models tends to vary for different soil heavy metals. Few studies have delved into a detailed examination of the similarities and differences in characteristic bands among different heavy metals. Additionally, there is limited investigation on strategies to minimise redundancy in characteristic bands and to extract the unique spectral responses of each heavy metal element. The improvement of this study is that the three–band spectral index combination of Pb, As, Mn and Zn was improved by using the developed RCBP and Catboost methods, which reduced the correlation between the characteristic bands of heavy metals and avoided interaction compared to previous studies (Table 3). The correlation between Fe and Pb, As, Mn and Zn was studied in terms of concentration and characteristic band, which supported the indirect inversion of heavy metal concentrations. Besides direct comparison between measured and predicted concentrations, the present study further conducted spatial correlation analysis and cluster analysis to evaluate the established inversion models for Pb, As, Mn, and Zn.

## 5. Conclusions

This study used RCBP to screen three-band spectral index combinations of four elements (Pb, Zn, Mn, and As) in entisols. We analysed the correlations between each element and Fe in terms of concentrations and characteristic bands, inverted the concentration of each element using the Catboost algorithm, and evaluated the effect of spectral inversion with the help of spatial distribution feature analysis. The conclusions of the study are as follows.

When each of the four elements was used as the target element to reduce the correlation between the characteristic bands of the target element and other elements, the extracted characteristic bands of Pb were 450, 466, 622, 1278, and 1530 nm, those of As were 622, 746, 930, 938, 1102, 1122, and 1274 nm, those of Zn were 450, 622, 630, and 1230 nm, and those of Mn were 1318, 1646, 1806, 2271, 2275, and 2383 nm. Based on combinations of the spectral indices of each element, the regression model of the inversion of the Zn concentration had the best fit, with *R*^2^ = 0.8786 for the test set, followed by the models for Mn (*R*^2^ = 0.8576), As (*R*^2^ = 0.7916) and Pb (*R*^2^ = 0.6022).

Fe concentrations had the strongest correlation with Mn concentration (r = 0.82), followed by Zn and As, while the weakest correlation was with Pb (r = 0.373). In terms of the characteristic bands, Fe concentrations were most strongly correlated with Pb in the 458, 654 and 1090 nm characteristic bands (r = 0.837). In the 458, 474, 654 and 1090 nm characteristic bands, Fe had the strongest correlation with Zn (r = 0.711). In the 458, 474, 522, 630, 654, 930, 1090, 1330 and 1694 nm characteristic bands, Fe has the highest correlation coefficient with Mn (r = 0.542). The correlation between Fe and As in the characteristic bands was weak (r < 0.303).

The distributions of the measured and spectral-predicted concentrations are consistent for all studied elements (As, Mn, Pb, and Zn), but the predicted concentrations for Pb and Zn exhibited some differences compared to the measured concentrations in local areas. In terms of spatial correlations, the predictions of As and Mn concentrations were good, and those of Pb and Zn were weaker. In the spatial clustering analysis, the predictions of As and Mn concentrations were better than those of Pb and Zn.

## Figures and Tables

**Figure 1 sensors-24-01492-f001:**
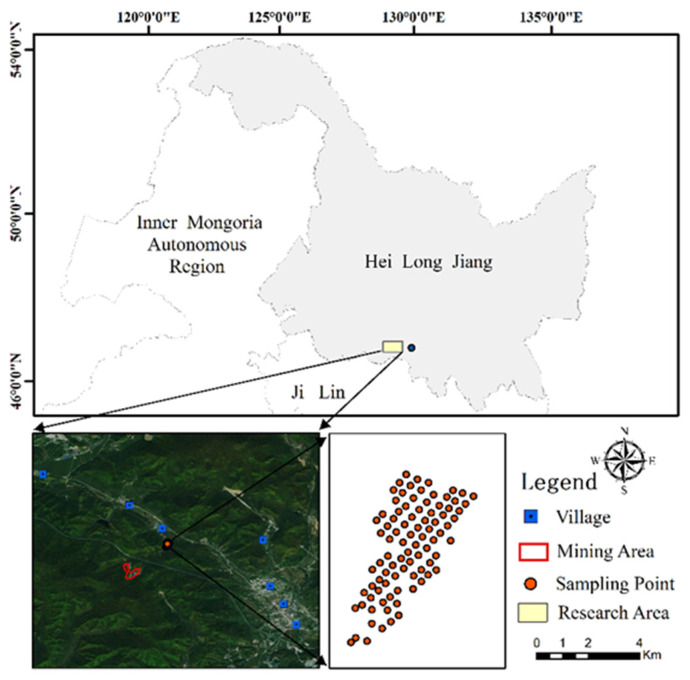
Location of the study area and sampling site distribution.

**Figure 2 sensors-24-01492-f002:**
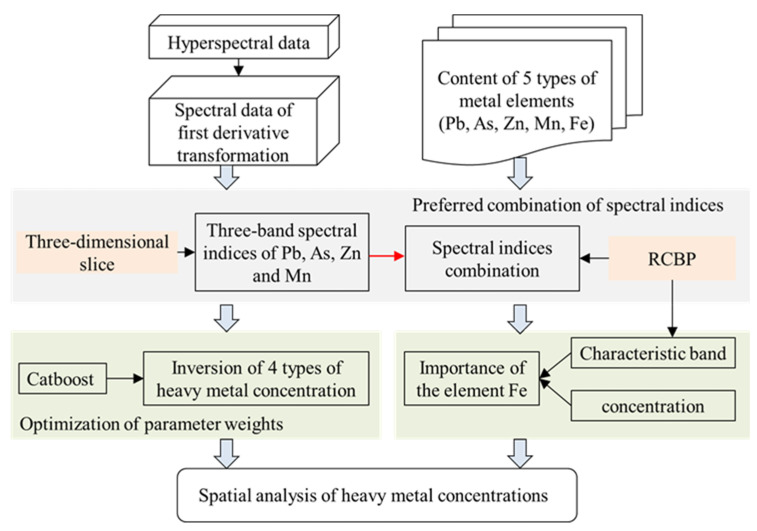
Flowchart of the analysis process.

**Figure 3 sensors-24-01492-f003:**
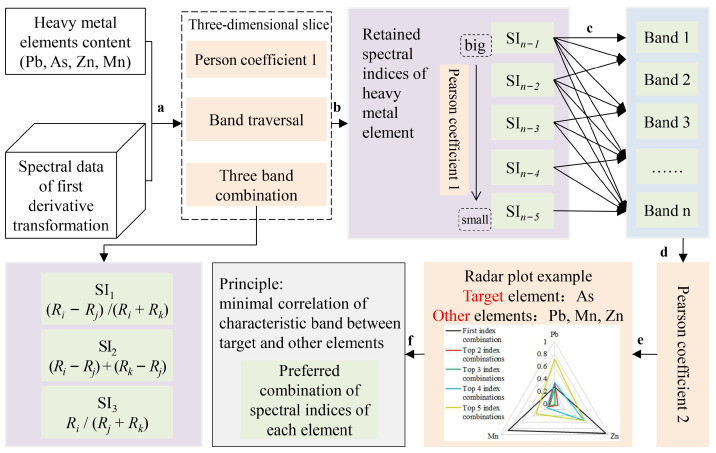
Preferred combination of spectral indices for heavy metal elements based on RCBP.

**Figure 4 sensors-24-01492-f004:**
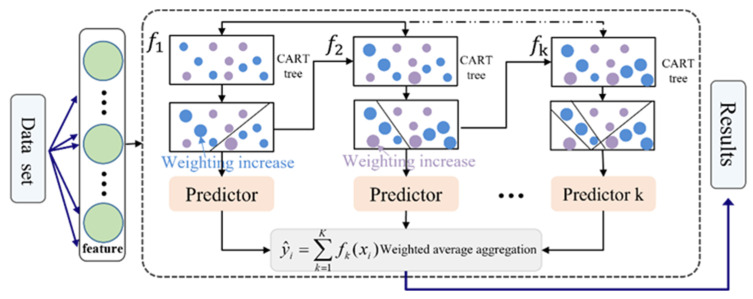
Schematic diagram of Catboost algorithm.

**Figure 5 sensors-24-01492-f005:**
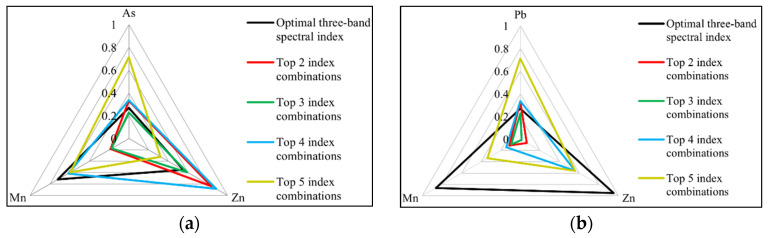
Radar maps of correlation coefficients between the characteristic bands of the target elements and other elements. (**a**) Target element = Pb, (**b**) Target element = As, (**c**) Target element = Zn, (**d**) Target element = Mn.

**Figure 6 sensors-24-01492-f006:**
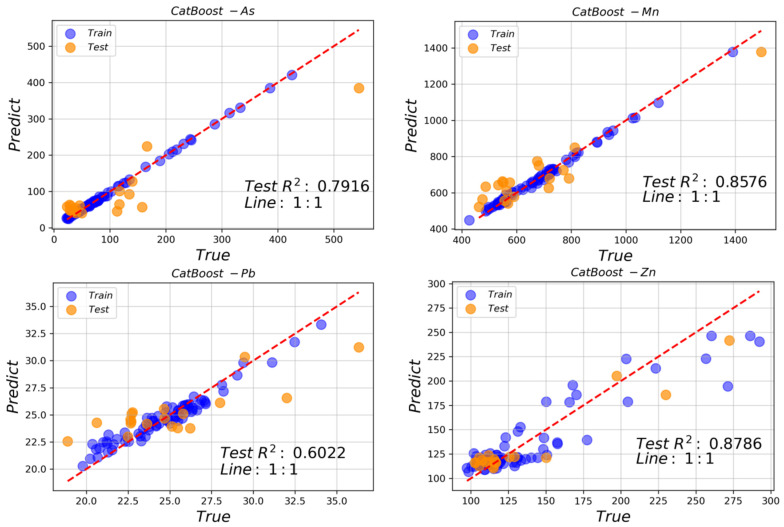
Comparison of the measured concentrations and those estimated by the Catboost algorithm for each heavy metal in the test set (As: *p*-value < 0.05; Mn: *p*-value < 0.01; Pb: *p*-value < 0.05; Zn: *p*-value < 0.05).

**Figure 7 sensors-24-01492-f007:**
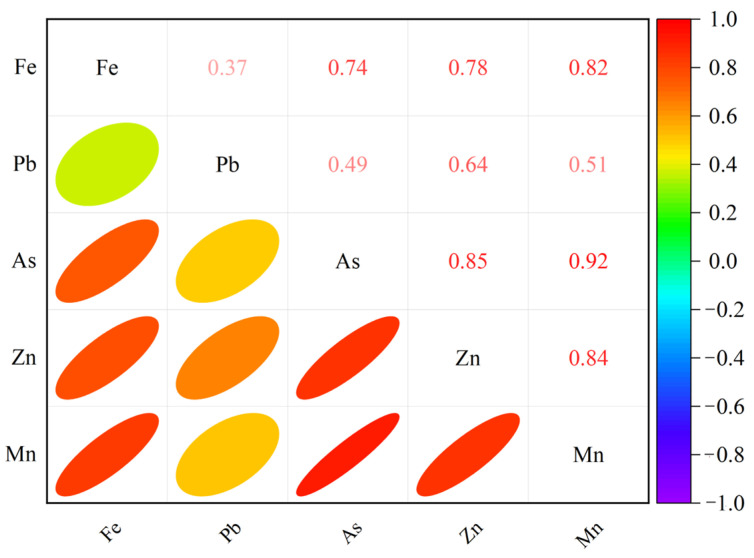
Pearson correlation matrix of elemental concentrations (Fe, Pb, As, Zn, and Mn) in soils from entisols in a metal mining-impacted area in China.

**Figure 8 sensors-24-01492-f008:**
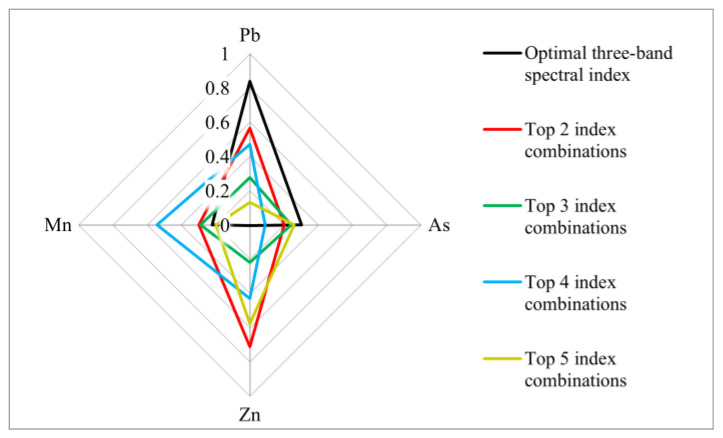
Correlations between characteristic bands of Fe and those of Pb, As, Zn, and Mn based on different spectral index combinations.

**Figure 9 sensors-24-01492-f009:**
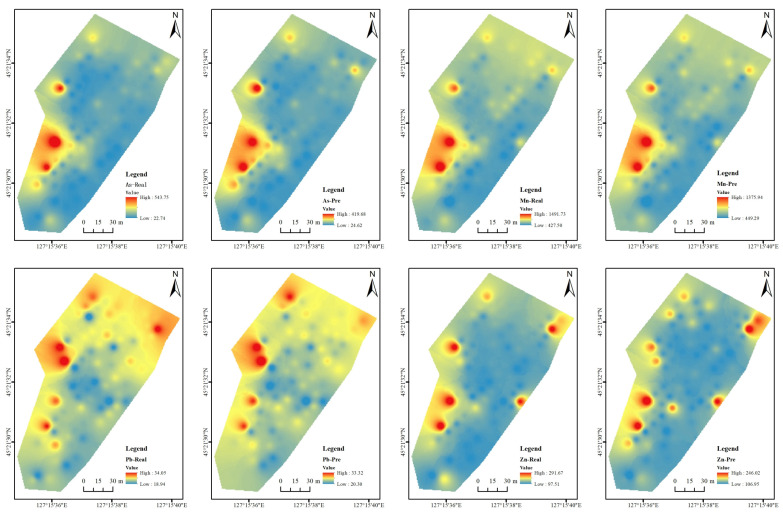
Spatial distributions of soil heavy metal contents (measured and predicted values).

**Figure 10 sensors-24-01492-f010:**
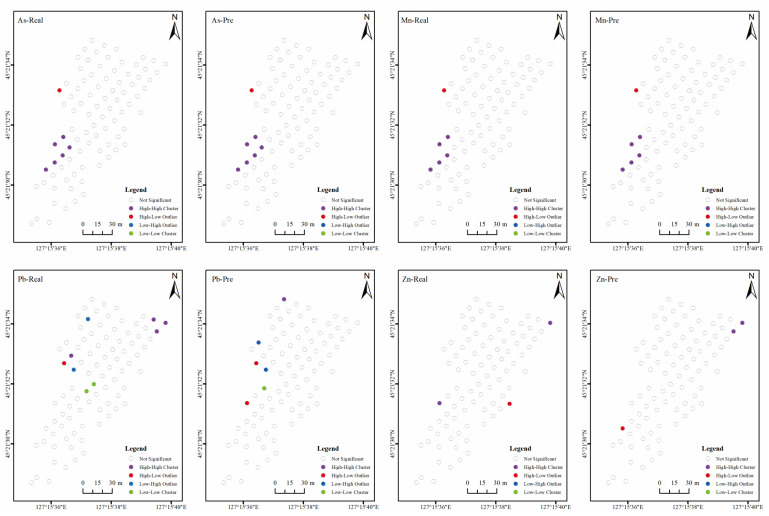
Spatial distributions of clusters of predicted and real concentrations of the four soil heavy metals.

**Table 1 sensors-24-01492-t001:** The top five groups of spectral indexes have good correlations with the five elemental contents.

Element	Spectral Indices	Range of *r* Values
Pb	① (R_FD1278_ − R_FD622_) + (R_FD1530_ − R_FD622_)	0.5893−0.5946
	② (R_FD466_ − R_FD622_) + (R_FD1530_ − R_FD622_)	
	③ (R_FD450_ − R_FD622_) + (R_FD1530_ − R_FD622_)	
	④ R_FD1278_/(R_FD534_ − R_FD1682_)	
	⑤ (R_FD1278_ − R_FD666_) + (R_FD1530_ − R_FD666_)	
Zn	① (R_FD450_ − R_FD622_) + (R_FD1230_ − R_FD622_)	0.6608−0.6744
	② (R_FD466_ − R_FD630_) + (R_FD1530_ − R_FD630_)	
	③ (R_FD462_ − R_FD622_) + (R_FD746_ − R_FD622_)	
	④ (R_FD450_ − R_FD622_) + (R_FD1530_ − R_FD622_)	
	⑤ (R_FD450_ − R_FD622_) + (R_FD1198_ − R_FD622_)	
Mn	① R_FD2271_/(R_FD2275_ + R_FD1806_)	0.6440−0.6595
	② (R_FD1646_ − R_FD2383_)/(R_FD1646_ + R_FD1318_)	
	③ (R_FD1806_ − R_FD2263_)/(R_FD1806_ + R_FD2275_)	
	④ (R_FD1094_ − R_FD1638_)/(R_FD1094_ + R_FD1106_)	
	⑤ R_FD1342_/(R_FD2275_ + R_FD1806_)	
As	① (R_FD1274_ − R_FD1102_)/(R_FD1274_ + R_FD930_)	0.6657−0.6756
	② (R_FD622_ − R_FD746_)/(R_FD622_ + R_FD1122_)	
	③ (R_FD930_ − R_FD938_)/(R_FD930_ + R_FD1274_)	
	④ (R_FD1274_ − R_FD1682_)/(R_FD1274_ + R_FD930_)	
	⑤ (R_FD622_ − R_FD738_)/(R_FD622_ + R_FD1122_)	
Fe	① R_FD458_/(R_FD1090_ + R_FD654_)	0.6077−0.6380
	② R_FD474_/(R_FD1090_ + R_FD654_)	
	③ (R_FD522_ − R_FD630_)/(R_FD522_ + R_FD934_)	
	④ (R_FD1694_ − R_FD930_)/(R_FD1694_ + R_FD1330_)	
	⑤ R_FD458_/(R_FD1310_ + R_FD654_)	

*p*-value < 0.001.

**Table 2 sensors-24-01492-t002:** Summary of the global Moran’s I values of the four heavy metal elements in soil.

Heavy Metal Element		Expected Index	Variance	*Z*-Score	*p*-Value	Moran’s *I*
As	Real	−0.0111	0.0048	3.6874	0.0002	0.2450
	Pre	−0.0111	0.0050	3.5956	0.0003	0.2440
Mn	Real	−0.0111	0.0050	3.7931	0.0001	0.2563
	Pre	−0.0111	0.0050	3.1619	0.0015	0.2120
Pb	Real	−0.0111	0.0054	2.8937	0.0038	0.2017
	Pre	−0.0111	0.0053	1.2835	0.1993	0.0821
Zn	Real	−0.0111	0.0051	2.0108	0.0443	0.1323
	Pre	−0.0111	0.0052	0.6522	0.5142	0.0358

**Table 3 sensors-24-01492-t003:** Selected spectral wavelengths in literature for estimating heavy metal concentrations.

Heavy Metals Studied	Heavy Metals Studied Selected Spectral Bands (nm)	Reference
Pb, Zn, Cu, Cd, Mn	800, 1300	[24]
Pb, Zn, Mn	500, 610	[51]
Cd, Cu, Pb, Zn, Ni, Mn, Cr, Co, Fe	538, 578, 630, 870, 1900, 2240, 2376	[52]
Al	480, 500, 565, 610, 680, 750, 1000, 1430, 1755, 1887, 1920, 1950, 2210, 2260	[9]
Cu	480, 500, 610, 750, 860, 1300, 1430, 1920, 2150, 2260	
Cr	480, 500, 610, 715, 750, 860, 1300, 1430, 1755, 1920, 1950	
Fe, Cu, Zn, Hg	486, 424, 1546, 1632, 1462, 1658, 1736, 1832, 1924, 2360	[53]
Cd	552, 698, 814, 1042, 1370, 1546, 1722, 1868, 2360, 2924	
Ni	552, 698, 814, 1042, 1332, 1546, 1722, 1868, 2360, 2924	
Pb	450, 466, 622, 1278, 1530	This research
As	622, 746, 930, 938, 1102, 1122, 1274	
Zn	450, 622, 630, 1230	
Mn	1318, 1646, 1806, 2271, 2275, 2383	

## Data Availability

Dataset available on request from the authors.

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
