# Peer review of "Estimating the Heavy Metal Contents in Entisols from a Mining Area Based on Improved Spectral Indices and Catboost"

_sensors, 2024, doi:10.3390/s24051492_

Round 1

Reviewer 1 Report

Comments and Suggestions for Authors

In this study, regarding the problem of redundant and unstable spectral features due to interaction effects among soil elements, the radar plots of characteristic-band Pearson coefficients were used to screen spectral index combinations. This is a carefully done study and the findings are of considerable interest, and can be published after answering the following questions:

1. In the Section of Methods, how many samples were for model training and test, respectively.

2. How to collect the soil samples was not explained. Please add the collection method (sampling depth? sampling interval?).

3. Please explain the rationale for constructing the three forms of spectral indices. Why were the three spectral bands combined in these forms to contruct the spectral indices?

4. The spectral indices were retained by traversing all possible triple-band combinations over the entire band range. Each spectrum measured by ASD field spectrometer exceeded 1000 bands, which means at least 108 calculations were needed to gain spectral index for one sample. Will the computational complexity affect the applicability of the method?

5. Please explain why the prediction of As and Mn element concentrations is effective, while the prediction of Pb and Zn element concentrations is poor?

Reviewer 2 Report

Comments and Suggestions for Authors

The article explores an intriguing topic of scientific significance; however, the structure of the manuscript requires significant improvement, and there is a need for a better presentation of results.

Comments on the Quality of English Language

L50-52. Revise to “Malley and Williams (1997) were the first to employ hyperspectral remote sensing…”.

L84. Revise to “absorption”.

L214. Define the acronym SI

L291. One bracket is missing.

Reviewer 3 Report

Comments and Suggestions for Authors

In this manuscript, the authors proposed a method which used RCBP to screen three-band spectral index combinations of different elements. The combination of spectral indices and their weights were optimized and the could be used for identification of heavy metals in soils. There are still some details need to be addressed in the manuscript, I listed as below.

1.     Although the novelty of this work is mainly the spectral processing method and feature band extraction method, the detection theory basis still need to be given. Why this signal could be related to heavy elements?

2.     Which kind of hyperspectral technique is used here for the heavy metal elements detection? The FieldSpec 4 Hi-Res portable ASD geophysical spectrometer is a SWIR resolution spectrometer which could be used in many fields. For your detection, is the system active or passive? Does it include a light source, and is it based on reflective collection?

3.     The authors mentioned the pXRF, ICP-MS and ICP-AES together when they mentioned the sample preparation. As I understand, they all can used for the elements’ measurement, and here they are used for reference and verification of the prepare results. But it’s not quite clear here. Better to separate the sample preparation and verification in two paragraphs.

4.     In Figure 2, the upper edges of the picture are missing. Some other figures also have tis problem, please check the figure edges.

5.     The equations of R2 and RMSE should be typed not insert as a figure, at least improve the resolution.

Comments on the Quality of English Language

No hyphens are needed in the manuscript when it doesn't appear at the end of the line.

Round 2

Reviewer 2 Report

Comments and Suggestions for Authors

"The authors improved the text, taking into account the comments, and made all necessary corrections.